# Extracapsular Tonsillectomy versus Intracapsular Tonsillotomy in Paediatric Patients with OSAS

**DOI:** 10.3390/jpm13050806

**Published:** 2023-05-08

**Authors:** Massimo Mesolella, Salvatore Allosso, Valentina Coronella, Eva Aurora Massimilla, Nicola Mansi, Giovanni Motta, Grazia Salerno, Gaetano Motta

**Affiliations:** 1Unit of Otorhinolaryngology, Department of Neuroscience, Reproductive Sciences and Dentistry, University Federico II of Naples, 80138 Napoli, Italy; 2Unit of Otorhinolaryngology, University Luigi Vanvitelli, 80138 Napoli, Italy; 3Otorhinolaryngology Unit, AORN Santobono-Pausilipon, 80112 Naples, Italy

**Keywords:** tonsillectomy, tonsillotomy, OSAS, surgery

## Abstract

Objective: The objective of our study was to compare our experience of intracapsular tonsillotomy performed with the help of a microdebrider usually used for adenoidectomy with results obtained from extracapsular surgery through dissection and from adenoidectomy in cases of people affected with OSAS, linked to adeno-tonsil hypertrophy, observed and treated in the last 5 years. Methods: 3127 children with adenotonsillar hyperplasia and OSAS-related clinical symptoms (aged between 3 and 12 years) underwent tonsillectomy and/or adenoidectomy. A total of 1069 patients (Group A) underwent intracapsular tonsillotomy, while 2058 patients (Group B) underwent extracapsular tonsillectomy, from January 2014 to June 2018. The parameters considered in order to evaluate the effectiveness of the two different surgery techniques taken into consideration were as follows: the presence of possible postoperative complications, represented mainly by pain and perioperative bleeding; the level of postoperative respiratory obstruction compared with the original obstruction through night pulse oximetry, performed 6 months before and after the surgery; tonsillar hypertrophy relapse in Group A and/or the presence of residues in Group B with clinical evaluation performed 1 month, 6 months, and 1 year after the surgery; and postoperative life quality, evaluated through submitting to parents the same survey proposed before the surgery 1 month, 6 months, and 1 year after the surgery. Results: Regardless of the technique used (extracapsular tonsillectomy or intracapsular tonsillotomy), there was a clear improvement in both the obstructive respiratory symptomatology and quality of life in both patient groups, as highlighted by the pulse oximetry and the OSA-18 survey submitted later. Conclusions: Intracapsular tonsillotomy surgery has improved in terms of a reduction in postoperative bleeding cases and pain reduction, with an earlier return to patients’ usual lifestyle. Lastly, using a microdebrider with the intracapsular technique seems to be particularly effective in removing most of the tonsillar lymphatic tissue, leaving only a thin border of pericapsular lymphoid tissue and preventing lymphoid tissue regrowth during one year of follow-up.

## 1. Introduction

Obstructive sleep apnoea syndrome (OSAS) is a breathing disorder that occurs during sleep and is characterised by a prolonged partial obstruction (hypopnoea) and/or by an intermittent complete obstruction (apnoea), which alters lung ventilation, gas respiratory exchanges, and sleep patterns [1,2].

OSAS preponderance in paediatric age groups can reach peaks at around 5–7% and the most common risk factor is the presence of adenotonsillar hypertrophy, followed by other independent pathogenetic factors such as obesity, craniofacial anomalies, neuromuscular pathologies, etc. [1,2].

Adenotonsillar hypertrophy represents the major cause of OSAS development in children without comorbidities, and thus, adenotonsillectomy is recommended as the “first line” in OSAS treatment in paediatric age groups [2,3,4].

Tonsillectomy is the most common otorhinolaryngologic surgery in paediatric age groups. The clinical methodology consists of the complete removal of the palatine tonsils, capsule included, followed by dissection, which is carried out inside the peritonsillar space. This is anatomically located between the tonsil capsule and the pharyngeal musculature; in this case, the procedure consists of total extracapsular tonsillectomy [3,4,5,6,7,8]. This surgical procedure, despite being routinely used, is weighed down with perioperative risks that, in spite of being less frequent, have to be taken into account, such as tonsillar hemorrhage [3,5].

Therefore, intracapsular tonsillotomy represents a surgical alternative to total extracapsular tonsillectomy in paediatric patients who are afflicted with OSAS. It is not possible to perform intracapsular tonsillectomy in cases of recurrent strep tonsillitis or PANDAS.

This procedure aims to minimise the postoperative morbidity of conventional surgery with equal functional results. The methodology consists of reducing the respiratory obstruction caused by tonsil hyperplasia through parenchyma volume reduction; this involves partial removal of the tonsillar tissue, leaving a minimal part of lymphoid tissue in order to “protect” the tonsil capsule and the underlying muscular tissue in the lateral oropharynx wall [4,5,6].

The objective of our study was to compare our experience of intracapsular tonsillotomy carried out using a microdebrider together with adenoidectomy (fundamental surgery stage in paediatric OSAS treatment), with results obtained from extracapsular tonsillectomy through dissection and adenoidectomy, with respect to patients affected with OSAS linked to adenotonsillar hypertrophy who were observed and treated in the last 5 years.

We proceeded with a comparison between the two surgery techniques, evaluating in particular their effectiveness related to the following:○OSAS clinical improvement after surgery;○the possible presence of early and/or late haemorrhagic complications;○postoperative pain monitoring. 

## 2. Materials and Methods

### 2.1. Case Study 

The study included 5000 patients aged between 3 and 12 years old, consecutively led to the otolaryngologic multispecialty department of the AORN (hospital of national relevance) “Santobono-Pausilipon” in Naples from January 2014 to June 2018, who were affected with OSAS linked to adenotonsillar hypertrophy. 

The study was conducted excluding patients with the following: neuro-muscular pathologies; major craniofacial anomalies, both isolated and linked to syndromes; obesity; recurring tonsil infections; and documented allergy history (skin prick test) with respect to common air allergens.

All patients underwent the following:

I.Anamnesis

This was conducted in order to evaluate the medical history of snoring, respiratory pauses during sleep, drowsiness and day-time hyperactivity, school performance, and behaviour. 

II.Survey OSA-18 specific for SBD (obstructive sleep-disordered breathing)

During the anamnesis we distributed a survey borrowed from the literature (OSA-18) to the patients’ parents. This aimed to define the quality-of-life (QoL) alteration in the patients, as the objects of the study, relative to the obstructive symptomatology they presented [9,10,11,12,13].

The survey consisted of 18 questions regarding 5 fields: sleep disorders, symptomatology, emotional impact, daily activities, and parents’ worries. Each of these was given a score from 1 to 7, with a final summed score that could vary from 18 to 126. 

 -A score below 60 implied a minor impact on the quality of life. -A score between 60 and 80 meant a moderate impact. -A score above 80 meant a major negative impact on the patient’s quality of life.

By calculating the average of the single values, we found the following:

 -1873 patients (37.46%) presented values < 60 (minor impact); -2408 patients (48.16%) presented values between 60 and 80 (moderate impact); -719 patients (14.38%) presented values between 80 and 105 (major impact).

III.Upper respiratory tract endoscopy:

All the patients underwent a complete otolaryngologic visit, on an outpatient basis; specifically, an upper respiratory tract evaluation was performed through the employment of an Olympus System^®^ flexible (Olympus Europa SE & Co. KG, Wendenstraße 20, 20097 Hamburg) video-endoscope (with a diameter of 2.7 mm for patients ranging from 3 to 6 years, and 3.5 mm for patients older than 6 years) equipped with a chip on the tip, connected to a camera that was able to decode the images collected. All the visits were observed in order to evaluate the pharynx anatomical district. The visits were performed without administering local and/or general anaesthetics.

In order to define the adenoid hypertrophy degree, we referred to the most commonly used classification system, devised by Parikh et al. [14] which identifies 4 obstruction degrees based on the existing relationship between the hypertrophic adenoids, vomer, soft palate, and torus tubarius:-First-degree obstruction—adenoid volume limited to the nasopharynx roof;-Second-degree obstruction—adenoids reach the upper half of the nasopharynx;-Third-degree obstruction—adenoid volume covers ¾ of the nasopharynx space;-Fourth-degree obstruction—adenoids completely obstruct the nasopharynx.

For tonsil hypertrophy diagnosis, the examination employed was the pharynx endoscopy. This employs the Mackenzie classification [14], which also identifies 4 degrees:-1st degree—intravelic tonsils, completely behind the front tonsillar pillar;-2nd degree—lightly hypertrophic tonsils, barely protruding from the front tonsillar pillar;-3rd degree—hypertrophic tonsils, occupying ¾ of the available space;-4th degree—extremely hypertrophic tonsils, completely obstructing the aero-digestive tract (kissing tonsils).

The procedure highlighted the following: ○1873 patients (37.46%) presented 1st–2nd degree adenotonsillar hypertrophy; ○3127 patients (62.54%) presented 3rd–4th degree adenotonsillar hypertrophy. 

Thus, following this first set of analyses, we selected 1873 patients (37.46%) with 1st–2nd degree adenotonsillar hypertrophy with a related score on the OSA18 survey highlighting a modest impact on quality of life (average score < 60). Together with a minor symptomatology, this ensured that the patients would only undergo the surgical procedure (nasal corticosteroids and ionised nasal irrigation with hypertonic and isotonic saline solutions) and would be excluded from the study.

In the 3127 remaining patients (62.54%), the rhinopharynx endoscopy highlighted a 3rd–4th degree lymphatic hypertrophy and all the patients had a survey score > 60. In particular, 2408 (48.16%) patients presented a score between 60 and 80 (moderate impact) and the remaining 719 (14.38%) presented a score between 80 and 105 (major impact), with an average score of 89.5. This last group was also selected to undergo night pulse oximetry in order to evaluate the OSAS degree and undergo surgery [14,15,16,17,18,19,20] (Table 1).

IV.Night pulse oximetry

The exam was carried out using a pulse oximeter, which allowed us to measure blood oxygen saturation (SpO2). The exam was carried out at the patient’s residence using a Gima Oxy 50 device, with result readings performed by medical personnel from the pneumology multispecialty department of AORN “Santobono-Pausilipon” the morning after the exams. The track was evaluated using the McGill Oximetry Scoring System (MOS), following the criteria set by Nixon et al. This system was devised as a screening method to select patients with lymphatic tissue hypertrophy and direct them to have an adenotonsillectomy. It consists of a tool diagnostic score, which, starting from the number of desaturations < 90% and the number of cluster desaturations, divides up the patients by desaturation degree. Desaturation is considered a saturation reduction of at least 4% relative to the basal value, and cluster desaturation describes the presence of at least 5 desaturations in a period of time between 10 and 30 min. Therefore, the examination yields positive results in the presence of the following:
○3 desaturation clusters or at least 3 desaturations < 90% (during the night recording) (Category 2, minor OSAS); ○4 desaturations < 85% (Category 3, moderate OSAS); ○4 desaturations < 80% (Category 4, major OSAS). 

It yields negative results in the absence of desaturation clusters and in the absence of desaturations < 90% [15,16,17].

In our case study:○1879 patients (60.09%) presented Category 3, moderate OSAS;○1248 patients (39.91%) presented Category 4, major OSAS.

All patients with 3rd–4th degree lymphatic hypertrophy thus presented moderate-major OSAS (Figure 1).

### 2.2. Inclusion and Exclusion Criteria

The 3127 patients selected for the study presented the following: ➢Age between 3 and 12 years;➢OSAS diagnosis following the indications in the addressing document for the security and pertinence of adenoidectomy and tonsillectomy surgeries, elaborated by the Italian Ministry of Health: [21]
-medical history of respiratory sleep disorders;-3rd–4th degree adenotonsillar hypertrophy;-positive dynamic night pulse oximetry (Category 3–4 following McGill Oximetry Scoring System);-OSA-18 survey with a result ≥ 60;


The excluded patients presented the following:○1st–2nd degree adenotonsillar hypertrophy;○Cranio-facial anomalies;○Neuromuscular pathologies;○Obesity;○Recurring adenotonsillar infections with over 2 episodes a year;○Documented allergy history (through skin-prick test). 

### 2.3. Preoperative Procedure

A total of 3127 patients were selected from the starting sample, suffering from moderate-major OSAS linked to adenotonsillar hypertrophy. All patients were between 3 and 12 years old, with an average of 5.21 years (with a standard deviation of ±2.29), of whom 1845 were male (59% of total patients) and 1282 were female (41% of total patients) (Table 2). 

The choice between the intracapsular and extracapsular technique was made casually, at a ratio of 1:2, for all patients who were consecutively led to our care during the study period.

Consequently, 1069 patients (Group A), corresponding to 34% of the total sample, underwent intracapsular tonsillotomy, and 2058 patients (Group B), corresponding to 66% of the total sample, underwent extracapsular tonsillectomy (Figure 2).

The patients were operated on by ENT surgeons with expertise relating to the structures involved.

### 2.4. Surgery Type

The technique employed for intracapsular tonsillotomy consisted of moving the patient into the Rose position, ensuring exposure of the lower pole of both tonsils, and also using lateral mouth openers in order to facilitate greater tonsil exposition. The tool employed for tonsillotomy was a microdebrider with a 4 mm tip set to oscillating mode, with a frequency between 1000 and 3000 rpm. Bipolar forceps were used for haemostasis, as the microdebrider’s tip lacked monopolar diathermy. As for the technique details, after having positioned a retractor at the level of the anterior tonsillar pillar, the tonsillotomy began, starting from the posterior tonsillar pillar. Taking care not to damage the posterior tonsillar pillar, we then proceeded with progressive excision of the lymphatic tissue, thus leaving only a minimal part at the location, with the role of protecting the capsule. We focused on avoiding damage to the uvula, the tonsillar capsule, and the tonsillar pillar.

The extracapsular tonsillectomy used for Group B was carried out “cold” through scalpel dissection, surgical scissors, an ENT elevator, and a snare. In the Rose position, a McIvor mouth opener was used to achieve adequate oropharynx exposition. The tonsillectomy consisted of the incision of the anterior tonsillar pillar with a tonsil scalpel. After having identified the tonsillar capsule, we proceeded with the dissection using an ENT elevator, from the superior pole to the inferior pole. Lastly, the tonsil was removed completely using the snare. After the dissection and the haemostasis control, the same procedure was applied for the excision of the contralateral tonsil. After the haemostasis, the surgery was completed with the traditional “noose” suture in polyglactin 910 (vicryl rapid^®^) or with bipolar forceps for cauterisation of possible bleeding points (selective bipolar coagulation in order to reduce thermic lesions to the surrounding mucous membrane and muscles to a minimum) or both (mixed technique). In both groups, adenoidectomy was also performed using the Laforce adenotome and/or Beckmann curette, and haemostasis was performed through bipolar diathermy.

### 2.5. Postoperative Results Evaluation 

The parameters considered in order to evaluate the effectiveness of the two different surgery techniques were as follows: I.The presence of possible postoperative complications, mainly represented by pain and perioperative bleeding;II.The degree of postoperative respiratory obstruction, comparing it with the preoperative value through pulse oximetry; carried out 6 months after surgery;III.Tonsillar hypertrophy relapse in Group A and/or the presence of residuals in Group B with clinical evaluation taking place 1 month, 6 months, and 1 year after surgery;IV.Postoperative quality of life assessed through providing the patient’s parents with the same survey provided before the surgery 1 month, 6 months, and 1 year after the surgery.

In order to understand the presence of possible postoperative complications, all patients were subjected to pain evaluation two days following surgery. Pain was evaluated through the Wong–Baker scale, consisting of 6 faces, from a smiling face, corresponding to “no pain”, to a crying face, corresponding to “worst pain possible”. The scale was administered to each child, asking them to point out the face corresponding to the pain they felt. Each choice corresponded to a number from 0 to 5. Usually, the term “ill-being” is used for children of 3 to 5 years old and the term “pain” is used for older children [22]. 

Relative to haemorrhagic complications, all the cases of early (<24 h) or late (>24 h) postoperative bleeding, in which the patient needed revision surgery with general anaesthesia, were considered. In the case of bleeding, we proceeded with diathermic coagulation of the bleeding vessels or tonsillar pillar suturing, applied with a curved needle, where a nylon or silk thread sutured the two pillars together. 

Evaluation of the postoperative respiratory obstruction degree was effectuated through a new pulse oximetry evaluation 6 months after the surgery. Lastly, the postoperative quality-of-life variations and the clinical evaluation of tonsillar hypertrophy relapse (in Group A) or the presence of surgical residues (in Group B) were evaluated with a follow-up that consisted of an otolaryngologist evaluation with upper respiratory tract endoscopy and OSA-18 survey administered to the parents after 1 month, 6 months, and 1 year from the surgery.

### 2.6. Data Analysis

Statistical analysis was performed through Microsoft Excel^®^, analysing the epidemiological variables of age and sex, with mean and standard deviation, and using the Z-test, with a Gaussian distribution for bleeding variables in the intracapsular tonsillectomy (Group A) vs. extracapsular tonsillotomy (Group B), and comparing the therapeutic failures observed with pulse oximetry in patients from Groups A and B. The t-test was used to verify the improvement in the quality-of-life investigated via the OSA-18 survey before the surgery and 1 month, 6 months, and 1 year after the surgery. The *p*-value was considered statistically significant for values < 0.05.

The study was approved by the institutional review board committee of Federico II University of Naples, Italy (2020/205682). Informed written consent from the patients was obtained.

## 3. Results

### 3.1. Surgery Comparative Analysis

The patients’ demographic characteristics (age, sex) did not present statistically significant differences. The 3127 patients’ mean age was 5.21 years, with a standard deviation of ±2.29. 

Of the whole sample, 59% was male and 41% was female. 

Of the 3127 patients included in the study:

●1069 patients, Group A (equal to 34% of the children) underwent adenoidectomy and intracapsular tonsillotomy. Of these:
-172 patients were treated in 2014;-180 patients were treated in 2015;-203 patients were treated in 2016;-249 patients were treated in 2017;-265 patients were treated in 2018.


●2058 patients, Group B (equal to 66% of the children) underwent adenoidectomy and extracapsular tonsillectomy. Of these:
-510 patients were treated in 2014;-423 patients were treated in 2015;-395 patients were treated in 2016;-409 patients were treated in 2017;-321 patients were treated in 2018.


The number of tonsillectomies, as observed in the sample, exhibits progressive growth (from 25% in 2014 to 45% in 2018), while at the same time the number of tonsillotomy surgeries is decreasing (from 75% in 2014 to 55% in 2018) (Table 3); (Figure 3).

### 3.2. Postoperative Morbidity

With respect to postoperative pain control, patients who underwent intracapsular tonsillotomy presented a Wong–Baker scale score lower than the patients who underwent extracapsular tonsillectomy. Specifically, in Group A, of the 1069 patients, 853 (79.79%) reported “it quite hurts”, corresponding to 2 on the Wong–Baker scale, and 216 (20.21%) reported “it hurts”, corresponding to 3 on the aforementioned scale. In Group B, of the 2058 patients, 1646 (79.88%) reported “really a lot of pain”, corresponding to 5 on the Wong–Baker scale, and 412 (20.02%) reported “a lot of pain”, corresponding to 4. 

In summary, the results show a significantly lower score in Group A compared to Group B. 

Moreover, the patients in Group A exhibited a lower number of days with respect to needing painkillers (an average of 2 days for Group A and 4 days for Group B). 

In both groups, the hospitalization lasted only one night. The return to a regular diet and to normal daily activities happened on the sixth day for tonsillotomy patients, and on the tenth day for tonsillectomy patients (Table 4). 

Haemorrhagic complications were present in 50 cases of the 3127 surgeries (Table 5).

In Group A, which comprised patients who underwent intracapsular tonsillotomy (1069 patients), haemorrhagic complications appeared in 5 patients; that is, 0.16% of the surgeries. All the cases exhibited primary haemorrhage (<24 h) and needed revision surgery under general anaesthesia.

In Group B, subjected to extracapsular tonsillectomy (2058 patients), the haemorrhagic event appeared in 45 patients; that is, in 1.44% of the surgeries. In 80% of the bleeding cases (37 cases) the haemorrhage was secondary, as it occurred in the 24 h following the surgery. There were fewer cases of bleeding (20% (8 cases)) for primary haemorrhage with revision surgery. 

Cases of postoperative bleeding in intracapsular tonsillotomy were observed in 0.16% of the cases compared to 1.5% for the classic total tonsillectomy. 

In brief, the results show that cases of bleeding in children who underwent tonsillotomy were significantly lower compared to those who underwent tonsillectomy (*p* < 0.001), Z= −3.63.

### 3.3. Results and Clinical-Instrumental Follow-Up of Group A Patients: Intracapsular Tonsillotomy 

Patients were re-evaluated through otolaryngologist evaluation with endoscopy of the upper respiratory tract and the OSA-18 survey 1 month, 6 months, and 1 year after the surgery. Moreover, after 6 months, patients underwent a new evaluation through pulse oximetry (Table 6). 

Upper respiratory tract re-evaluation through rhinopharynx endoscopy: In patients who underwent intracapsular tonsillotomy, one month after the surgery, the evaluation highlighted the presence of tonsillar lymphoid tissue residuals (10 % of the original tonsillar parenchyma). This condition was present and stable even at later re-evaluations 6 months and 1 year after the surgery; in no case was hypertrophied residual lymphatic tissue observed, thus, no revision surgery was needed (Table 6).

OSA-18 survey: The results highlight a significant score reduction compared to the preoperative survey, where the average score for all patients was 89.5. We witnessed a progressive reduction in the symptomatology gravity and an improvement of quality of life since the first postoperative month (Table 6):
○541 patients, OSA-18: score up to 50 (minor impact of the pathology on the quality of life) (51% of cases);○528 patients, OSA-18: score between 50 and 85 (moderate impact of the pathology on the quality of life) (49% of cases).

However, after 6 months: 

○20 patients scored up to 60 (moderate impact of the pathology on the quality of life) (1.87% of the cases);○626 patients scored up to 40 (normal quality of life) (58.5% of the cases);○423 patients scored up to 30 (normal quality of life) (39.5% of the cases).

As evidenced by the reported data, in a small percentage of patients (1.87%, i.e., 20 patients), the quality-of-life survey still presented a score of 60, indicating daily ill-being in the patient. In these cases, we proceeded with a re-evaluation of the factors that could have influenced the surgery outcome.

After 1 year, the remaining 1049 patients were evaluated, of whom:
○846 patients presented a score up to 30 (normal quality of life) (80.6% of the cases); ○203 patients presented a score up to 18 (normal quality of life) (19.4% of the cases) (Figure 4).

Night pulse oximetry: The effectiveness of reducing the night obstructive symptomatology was evaluated through pulse oximetry 6 months after the surgery.

The results show a clinical statistically significant improvement as the exam became negative for 1049 (98.13%) patients; while it was positive for 20 patients (1.87%), with no observed improvement on the OSA-18 survey (Table 7; Figure 5).

The 20 patients (1.87%) who had positive pulse oximetry results and an OSA-18 score >60 (score indicating the ill-being of the patient) were clinically re-evaluated through multidisciplinary collaboration, aiming to check for the possible presence of comorbidities. These patients exited the study and followed the appropriate therapeutic path at the AORN Santobono-Pausilipon sleep treatment centre.

### 3.4. Results and Clinical and Instrumental Follow-up of Group B patients: Extracapsular Tonsillectomy

In this case, patients were re-evaluated by an otolaryngologist through upper respiratory tract endoscopy and OSA-18 survey 1 month, 6 months, and 1 year after the surgery, and a new pulse oximetry evaluation was undertaken after 6 months (Table 8). 

Upper respiratory tract re-evaluation through rhinopharynx endoscopy: Patients who underwent extracapsular tonsillectomy showed no sign of lymphoid tissue residuals one month after the surgery. The results were stable even at the next re-evaluations, which were carried out 6 months and 1 year after the surgery (Table 8).

The OSA-18 survey: one month after the surgery, we registered a rapid quality-of-life improvement, and progressive symptom reduction up to one year later (Table 8). 

At 1 month:○1198 patients had a score up to 40 (normal quality of life) (58% of the cases); ○860 patients had a score up to 60 (moderate pathology impact on the quality of life) (42% of the cases). 

At 6 months: ○36 patients had a score up to 60 (moderate pathology impact on the quality of life) (1.75% of the cases); ○1060 patients had a score up to 40 (normal quality of life) (51,5% of the cases);○962 patients had a score up to 30 (normal quality of life) (46.7% of the cases). 

As highlighted by the data, for a small percentage of the patients (1.75%, corresponding to 36 patients), the survey still showed a score of 60, indicative of the patient’s daily ill-being. In this case, we proceeded with a re-evaluation of the factors that could have influenced the surgical outcome.

After 1 year, 2022 patients were evaluated, of whom: ○1456 patients had a score up to 18 (normal quality of life) (71% of the cases);○566 patients had a score up to 30 (normal quality of life) (27.5% of the cases) (Figure 6).

Observing the results (Figure 7), it appears clear that following the extracapsular tonsillectomy surgery, patients showed a better quality of life improvement in the first month compared to patients who underwent tonsillotomy; however, after evaluating the next follow-ups, the data overlap and there are no significant differences (*t*-test = 0.44 with *p* = 0.03).

Pulse oximetry: The effectiveness of reducing the night obstructive symptomatology was evaluated through pulse oximetry 6 months after the surgery.

The results highlight a statistically significant clinical improvement, as the exam result became negative for 2022 (98.25%) patients; while it was still positive in 36 patients (1.75%), with no improvement in the OSA-18 survey (Table 9; Figure 8).

The 36 patients (1.75%) with positive pulse oximetry OSA-18 survey > 60 (score indicative of patient ill-being), who were clinically re-evaluated through multidisciplinary collaboration in order to check for the possible presence of comorbidities, exited the study and followed the appropriate therapeutic path at the AORN Santobono-Pausilipon sleep treatment centre (Table 9 and Table 10).

### 3.5. Cases of Postoperative Therapeutic Failure

In brief, according to our results, cases of therapeutic failure in intracapsular tonsillotomy were observed in 0.46% of the surgeries, compared to 1.15% for classic total tonsillectomy, as revealed by the pulse oximetry effectuated 6 months after the surgery (Table 10 and Table 11). The results show no statistically significant differences between the two techniques (test Z = 0.24 with *p* = 0.20).

Thus, 56 patients in our clinical records (1.79%) (36 from Group A (0.64%) and 20 from Group B (1.15%)), having obtained no benefit from the surgical therapy, needed to follow the diagnostic–therapeutic path with aid from other specialists (allergist, pulmonologist, odontologist, and phoniatrician) in order to highlight the possible presence of comorbidities that could have conditioned and altered the surgery outcome.

## 4. Discussion

The data observed thus far in the medical literature allow us to define intracapsular tonsillotomy as a valid therapeutic alternative to classic tonsillectomy in paediatric patients suffering from OSAS, with lower perioperative morbidity and better quality of life [4,6,9,10,11,23,24,25,26].

The two main benefits the medical literature attribute to this technique are postoperative pain and bleeding reduction [4,9,10,11,24,25]. The first is due to the lower traumatism, linked to the lack of tonsillar capsule removal and, thus, preservation of the constrictor muscle [5]. Reduced haemorrhagic events are legitimated because the intracapsular resection may cause damage to the blood vessels of lesser gravity compared to the muscle fascia [27,28]. Generally speaking, postoperative haemorrhage cases in intracapsular tonsillotomy are observed in less than 1% of cases, compared to 1–6% of cases for classic total tonsillectomy [29]. The data are upheld by the medical literature and agree with our study results. In particular, a retrospective study by Solares et al. observed a rate of postoperative haemorrhage in intracapsular tonsillotomy of 0.7% [28]. Analogously, Schmidt et al. reported postoperative haemorrhage values of 1.1% in intracapsular tonsillotomy and 3.4% in total tonsillectomy [26,30]. Chang et al. reported in a meta-analysis a significant statistical difference in the risk of perioperative bleeding, favouring the technique with the microdebrider (0.2% in intracapsular e 2.9% in extracapsular) [30]. In extracapsular dissection, muscle fibres of the palatoglossus or palatopharyngeus can sometimes be accidentally injured during cutting or simply by pulling the tonsil in the dissection. This does not occur in intracapsular dissection [26,28,29,30].

The current scientific evidence, however, not only confirms the postoperative morbidity reduction of the intracapsular technique compared to the conventional technique, but also highlights the substantial comparability of tonsillotomy compared to tonsillectomy in reducing the obstructive symptomatology and curing OSAS. The medical literature confirms the effectiveness of intracapsular tonsillotomy in reducing the AHI value and curing OSA, and in studies that compared the results of intracapsular and extracapsular techniques, both were deemed effective, lacking any statistically significant difference [31,32,33,34,35,36,37,38,39,40,41,42]. During our study, effectiveness was evaluated through night pulse oximetry [24,32,37] and the l’OSA-18 survey [12,13]. The results confirm a clinical improvement at 6 months after the surgery for both the techniques in the absence of significant statistical differences. In particular, the OSA-18 survey proved to be a valid tool for measuring subjective quality of life in paediatric patients afflicted by OSAS. In the medical literature, a recent and well-known meta-analysis (Gorman et al.) compared the scores obtained through the OSA-18 survey in patients who had undergone tonsillotomy and tonsillectomy and observed no significant statistical differences between the two patient groups [12]; this is in agreement with the results described in our study. 

A negative aspect that nevertheless appears in the medical literature regarding subtotal excision is the possibility of lymphatic tissue residual regrowth, which can lead to a relapse of the obstructive pathology [33,34,35,36], especially in the youngest patients, in recurring infections of the upper respiratory tract if they coexist with an allergic condition. The symptomatic regrowth of tonsillar tissue in tonsillotomy with a microdebrider is, however, described in the medical literature in less of 1% of cases [28], and we did not observe any cases of tonsillar lymphatic hypertrophy relapse requiring revision surgery.

## 5. Conclusions

Adenotonsillectomy is the first therapeutic choice in paediatric-age patients with an OSAS diagnosis linked to adenotonsillar hypertrophy. This surgery allows an improvement in sleep respiratory disorders, in the grand majority of cases, and remains the gold-standard for paediatric OSAS treatment [22,24,38,41,42,43,44,45]. With the aim of minimising surgical morbidity, with equal functional results, intracapsular tonsillotomy was introduced in our clinical procedure and carried out with the microdebrider [4,23,39,40]. Our results, in particular, show that regardless of the technique used (extracapsular tonsillectomy or intracapsular tonsillotomy), there was a tangible improvement in the obstructive respiratory symptomatology and of the quality of life in both the patient groups, as highlighted by the follow-up OSA-18 survey and the pulse oximetry results [9,10,11,12,13,17,25,32,37]. However, the study showed that intracapsular tonsillotomy obtained advantages in terms of the number of postoperative in situ bleeding reductions and pain reductions, with a faster return to daily habits [9,10,11,25,30]. The number of bleeding cases in our results have progressively decreased in the last 5 years, probably due to the increasing employment of intracapsular tonsillotomy compared to previous years. The use of a microdebrider in the intracapsular technique seems to be particularly effective in removing most of the tonsillar lymphatic tissue, leaving only a thin border of overcapsular lymphoid tissue, preventing lymphoid tissue regrowth during the follow-up, with clinical benefits for the patient (in fact, our clinical records did not present cases of relapse) [6,9,10,11,25]. However, not all the patients in our study obtained complete improvement from the surgery, regardless of the employed technique, as a small percentage of patients manifested only a modest clinical improvement. In these patients, however, the diagnostic procedure was carried out in order to treat pathological conditions besides the lymphatic tissue hypertrophy that contributed to the persistence of sleep respiratory disorders [24,32,39].

In conclusion, in this study, we highlighted that intracapsular tonsillotomy was effective in paediatric OSAS treatment, with less postoperative morbidity, in particular, and a tangible improvement in the obstructive respiratory symptomatology and the patients’ quality of life, with an absence of lymphatic tissue regrowth after up to 1 year of follow-up.

## Figures and Tables

**Figure 1 jpm-13-00806-f001:**
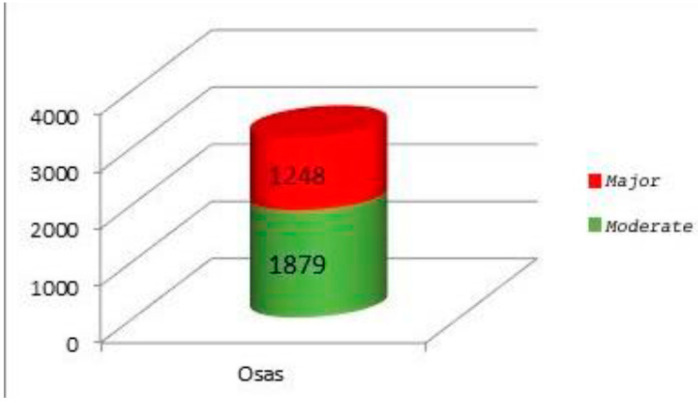
Preoperative pulse oximetry.

**Figure 2 jpm-13-00806-f002:**
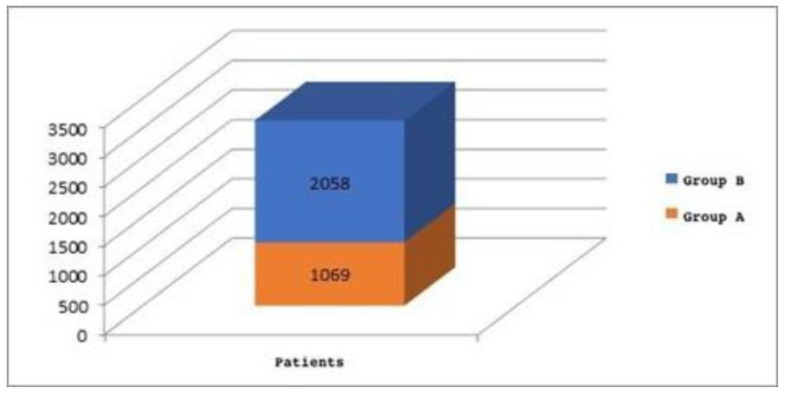
Group A: patients underwent intracapsular tonsillotomy; Group B: patients underwent extracapsular tonsillectomy.

**Figure 3 jpm-13-00806-f003:**
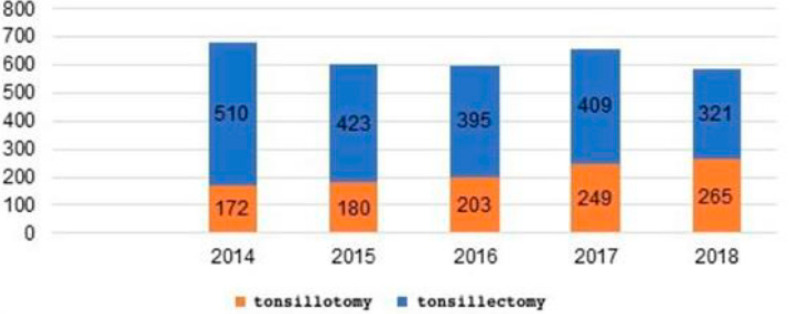
Tonsillotomy vs. tonsillectomy comparative analysis during the period 2014–2018.

**Figure 4 jpm-13-00806-f004:**
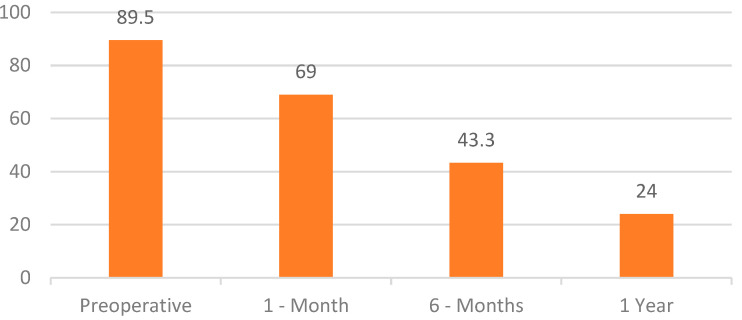
OSA-18 survey scores for intracapsular tonsillotomy.

**Figure 5 jpm-13-00806-f005:**
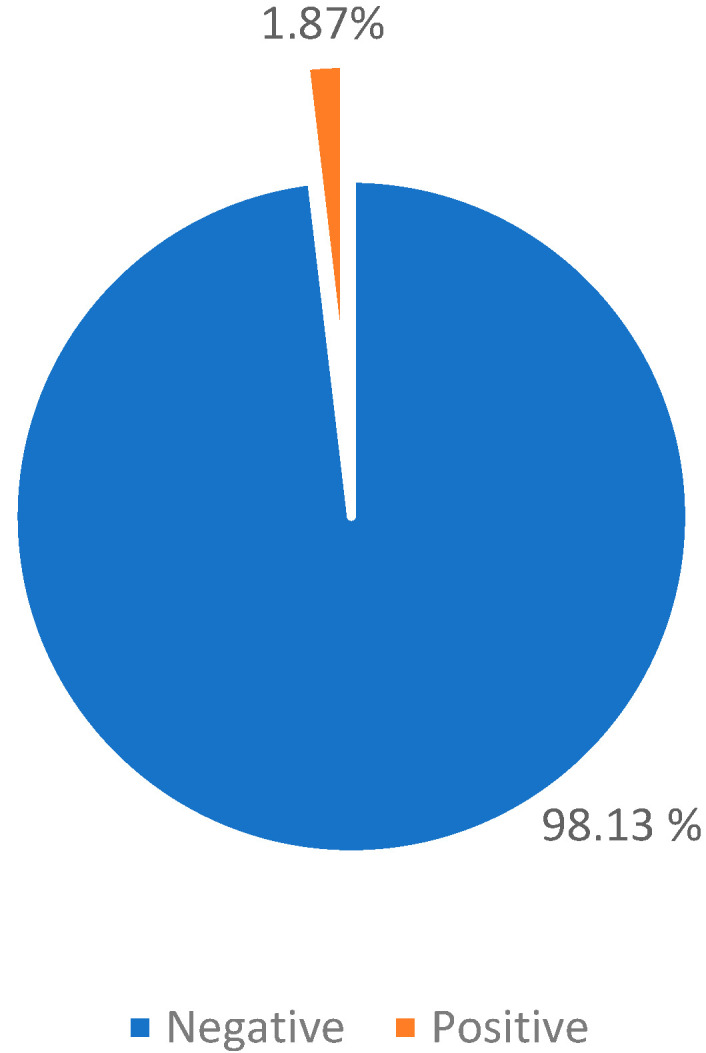
Pulse oximetry after tonsillotomy.

**Figure 6 jpm-13-00806-f006:**
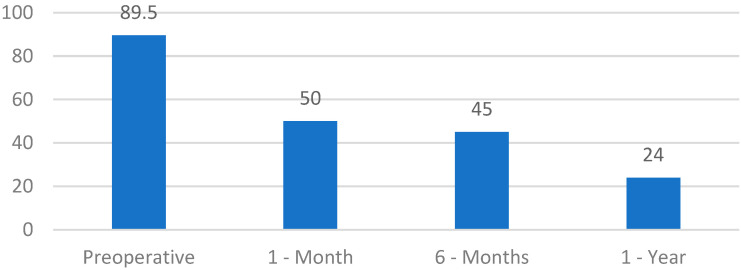
OSA-18 survey scores for extracapsular tonsillectomy.

**Figure 7 jpm-13-00806-f007:**
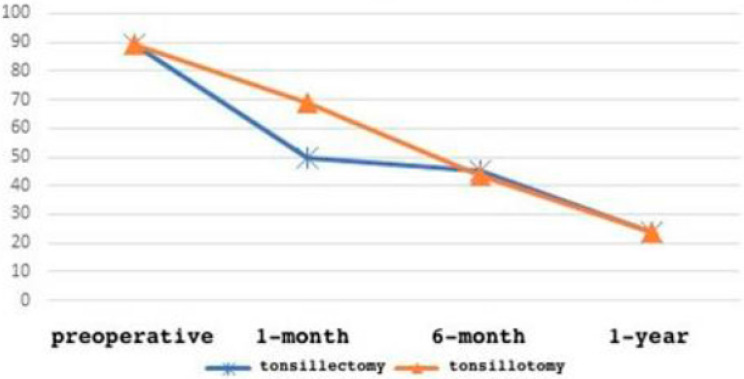
OSA-18 results comparative analysis.

**Figure 8 jpm-13-00806-f008:**
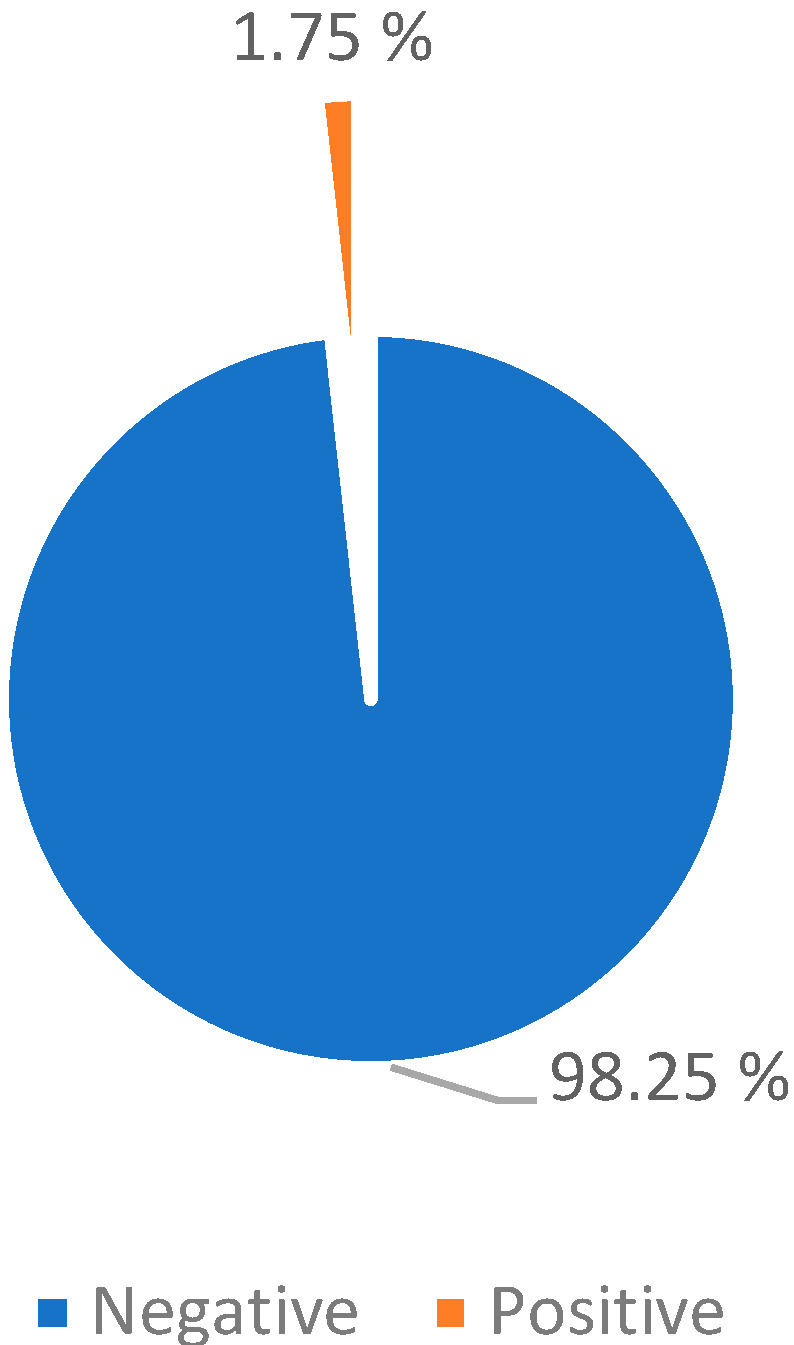
Post-tonsillectomy pulse oximetry.

**Table 1 jpm-13-00806-t001:** Case study.

Sample 5000 Patients	
3127 patients (62.54%) with:	1873 patients (37.46%) with:
3st–4nd degree adenotonsillar hypertrophy	1st–2nd degree adenotonsillar hypertrophy
OSA-18 survey score > 60	OSA-18 survey score > 60

**Table 2 jpm-13-00806-t002:** Sample statistical data.

**Average Age (Years)**	5.21
**St. Dev.**	±2.29
**% Males**	59%
**% Females**	41%

**Table 3 jpm-13-00806-t003:** Tonsillotomy (Group A) vs. tonsillectomy (Group B) comparative analysis in the period 2014–2018.

	Total Surgeries	Group A	%	Group B	%
2014	682	172	25%	510	75%
2015	603	180	30%	423	70%
2016	598	203	34%	395	66%
2017	658	249	38%	409	62%
2018	586	265	45%	321	55%
Total	3127	1069	34%	2058	66%

**Table 4 jpm-13-00806-t004:** Postoperative pain evaluation.

	Group A	Group B
Wong-Baker Scale	79.99% (2) 20.21% (3)	79.88% (5) 20.02% (4)
Days of paracetamol administering	2 days	4 days
Hospitalization	1 night	1 night
Return to solid diet	6 days	10 days
Return to daily activities	7 days	10 days

**Table 5 jpm-13-00806-t005:** Comparative analysis of post-tonsillotomy bleeding vs. post-tonsillectomy bleeding in the period 2014–2018.

	Total	Group A	% ^1^	Group B	% ^1^
Number of surgeries	3127	1069	34.19%	2058	65.80%
Number of post-operative bleedings	50	5	0.16%	45	1.44%

^1^: The % were calculated on the total of the surgeries.

**Table 6 jpm-13-00806-t006:** Group A: Patient comparison pre- and postsurgery.

	Preoperative	1 Month Follow-up	6 Months Follow-up	1 Year Follow-up
Rinopharynx endoscopy	100% of patients presented 3rd–4th degree adenotonsillar hyperplasia	Tonsillar lymphoid tissue residuals; Absence of adenoid lymphatic tissue	Clinical stability	Clinical stability
OSA-18	Average value 89.5	Average value 69.0	Average value 43.3	Average value 24.0
Pulse oximetry	Positive in 1069 patients (100%)		Negative in 1049 patients (98.13%) Positive in 20 patients (1.87%)	45

**Table 7 jpm-13-00806-t007:** Pulse oximetry before and after tonsillotomy.

	Peoperartive	Postoperative
Positive pulse oximetry	1069 patients	20 patients
% positive yielding pulse oximetry	100%	1.87%

**Table 8 jpm-13-00806-t008:** Group B: Patient comparison pre- and postsurgery.

	Preoperative	1 Month Follow-up	6 Months Follow-up	1 Year Follow-up
Rinopharynx endoscopy	3rd–4th degree adenotonsillar hyperplasia in 100% of the patients	Adenoid and tonsillar lymphoid tissue absence	Clinical stability	Clinical stability
OSA-18	Average value 89.5	Average value 50.0	Average value 46.7	Average value 27.5
Pulse oximetry	Positive in 2058 patients (100%)		Negative in 2022 patients (98.25%) Positive in 36 patients (1.75%)	45

**Table 9 jpm-13-00806-t009:** Preoperative and postoperative pulse oximetry.

	Peoperartive	Postoperative
Positive pulse oximetry	2058 patients	36 patients
% positive yielding pulse oximetry	100%	1.75%

**Table 10 jpm-13-00806-t010:** Postoperative pulse oximetry in Group A and Group B.

	Total	% ^1^	Tonsillotomy (Group A)	% ^1^	Tonsillectomy (Group B)	
Total surgeries	3127	100.00%	1069	34.19	2058	65.81%
Improved	3071	98.21%	1049	33.55%	2022	64.66%
Worsened	56	1.79%	20	0.64%	36	1.15%

^1^: the % were calculated on the total of the surgeries.

**Table 11 jpm-13-00806-t011:** Cases of postoperative therapeutic failure.

	Total	Tonsillotomy	% ^1^	Tonsillectomy	% ^1^
Surgeries	3127	1069	34.19%	2058	65.81%
Postoperative failures	56	20	0.64%	36	1.15%

^1^: the % were calculated on the total of the surgeries.

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
