# Peer review of "Extracapsular Tonsillectomy versus Intracapsular Tonsillotomy in Paediatric Patients with OSAS"

_jpm, 2023, doi:10.3390/jpm13050806_

Round 1
Reviewer 1 Report
This article summarized the surgical results of extracapsular tonsillectomy versus intracapsular tonsillotomy in total 3127 pediatric patients. Large sample size surely make the conclusion more convincing. However, there are several issues should be adressed and improved.
1. It is not clear how the grouping was done. Is there any selection bias in grouping? Suggest re-write Line 195-196. I think there's a typo in line 199: the percentage in group B should be 66%.
2. Were all surgeries performed by the same surgeon? For such large sample size, it is reasonable that those surgeries were performed by different surgeon, but should be elaborated in the text.
3. Several typo or wrong description should be corrected: Line 199: the percentage in group B should be 66%; Line 263-264: group A should be "intracapsular tonsillotomy"; Line 293-294: the description is imcompatible with results data; Figure 7 and 8 are imcompatible with results data shown in Table 8.
I suggest the author to proofread again and make sure all data presented in the paper is correct.
4. In group A, 20 patient resulted positive to pulse oximetry. Are these 20 patients exactly those who had OSA-18 score >60? Same question for group B.
5. All the decimal points should be "." but not ",".
6. Several figures and tables are not informative and unnecessary. Several fingures and tables should be combined and refined to highlight the important results.
7. Table 10 and 11 are almost duplication. In addition, I think it is more meaningful and reasonable to calculate the failed percentage on the case number of each group, but not the total 3172 surgeries.
8. I suggest add some interesting discussion, such as: Why do secondary bleeding only happened in extracapsular tonsillectomy? What is the actual reason of surgical failure?
Author Response
1 REVISORE:
This article summarized the surgical results of extracapsular tonsillectomy versus intracapsular tonsillotomy in total 3127 pediatric patients. Large sample size surely make the conclusion more convincing. However, there are several issues should be adressed and improved.
- It is not clear how the grouping was done. Is there any selection bias in grouping? Suggest re-write Line 195-196. I think there's a typo in line 199: the percentage in group B should be 66%.
The patients were randomly divided into the two groups, in a ratio of 1:2. The percentage in group B is 66%. Thanxs.
- Were all surgeries performed by the same surgeon? For such large sample size, it is reasonable that those surgeries were performed by different surgeon, but should be elaborated in the text.
The patients were operated on by ENT surgeons belonging to the structures involved-
- Several typo or wrong description should be corrected: Line 199: the percentage in group B should be 66%; Line 263-264: group A should be "intracapsular tonsillotomy"; Line 293-294: the description is imcompatible with results data; Figure 7 and 8 are imcompatible with results data shown in Table 8. I suggest the author to proofread again and make sure all data presented in the paper is correct.
I have correct.
- In group A, 20 patient resulted positive to pulse oximetry. Are these 20 patients exactly those who had OSA-18 score >60? Same question for group B.
Yes. The are the same
- All the decimal points should be "." but not ",".
I corrected and sent the work for English corrections.
- Several figures and tables are not informative and unnecessary. Several fingures and tables should be combined and refined to highlight the important results.
- Table 10 and 11 are almost duplication. In addition, I think it is more meaningful and reasonable to calculate the failed percentage on the case number of each group, but not the total 3172 surgeries.
We have chosen to keep the two tables separate because: table 10 refers to the pulse oximetry values by comparing the pre- and post-operative results in the two groups; table 11 refers to cases of therapeutic failure by comparing the cases out of the total and then specifically the cases in the two groups treated.
- I suggest add some interesting discussion, such as: Why do secondary bleeding only happened in extracapsular tonsillectomy? What is the actual reason of surgical failure?
In extracapsular dissection, muscle fibers of the palatoglossus or palatopharyngeus can sometimes be accidentally injured in the cut or simply by pulling the tonsil in the dissection. This occurrence does not occur in intracapsular dissection. (Line 481)

Reviewer 2 Report
Overall: This is a very important topic, as there remains controversy as to which is the ideal population for intracapsullar tonsillotomy and whether there are differences in outcomes in terms of pain and bleeding as compared to extra capsular dissection. This paper studies a large group of patients who underwent one of either of these procedures and compares outcomes.
While the paper is well written and does follow a large group of patients, what makes its results difficult to interpret and what is worth exploring more in the discussion is that this is not a randomized trial- the populations that underwent each procedure are likely different in some ways which may confound the results. How was it decided which procedure each patient received?
After addition of this point, I would support publication of this manuscript.
Abstract: Clearly written.
Introduction:
Again this is well written, although I do think it would be prudent to mention the types of patients for which intracapsular tonsillotomy is not appropriate (for example, recurrent strep or PANDAS in which all tonsillar tissue should be removed).
Methods:
The section is written well as well.
-What does it mean that decision for intracapsular tonsilloctomy vs tonsillectomy was made “casually” (page 6 line 195-6)? This suggests this was not random, which is important to mention and further explore, as this is not a randomized study. What factors went into deciding which treatment each patient received?
-Certainly a strength of the study is the size with 5000 patients.
Discussion:
-This section is concisely written and situates the current study well within the existing literature.
-There may be a few additional points worth adding.
-For example, it may be worth mentioning that patients with recurrent tonsillitis were excluded- these patients are not appropriate for tonsillotomy and may have higher rates of bleeding given recurrent inflammation around the tonsils themselves. Another large limitation here is age- one class of patients for which surgeons do strongly consider tonsillotomy is for is for children less then 3, and these appear to not be included in this study.
-After further explanation in methods as how patients were chosen for tonsillotomy vs. tonsillectomy, it would be worth exploring here what the implications are of this process (is there any possible selection bias in each group?)
-This study is also quite interesting in that both techniques here are using “cold” technique and no cautery- the intracapsular group had microdebrider and then the extra capsular had snare technique. It may be worth noting that their results may also be a function of the lack of cautery used in either group as well.
-A few wording revisions below:
-Page 15 line 459- “till now” is informal. Would change to “thus far”
-Page 15 line 463-“, is thanks so” is also informal- would change to “is likely due to”
-Page 16 line 497- using contractions is generally frowned upon in formal writing. Would change “didn’t” to “did not”
Formatting-
*The references do not appear in order in the text (meaning the #s for references do not follow 1,2,3 etc)-certainly up to journal whether this is formatted per their liking.
Author Response
Overall: This is a very important topic, as there remains controversy as to which is the ideal population for intracapsullar tonsillotomy and whether there are differences in outcomes in terms of pain and bleeding as compared to extra capsular dissection. This paper studies a large group of patients who underwent one of either of these procedures and compares outcomes.
While the paper is well written and does follow a large group of patients, what makes its results difficult to interpret and what is worth exploring more in the discussion is that this is not a randomized trial- the populations that underwent each procedure are likely different in some ways which may confound the results. How was it decided which procedure each patient received?
The patients were randomly divided into the two groups, in a ratio of 1:2.
After addition of this point, I would support publication of this manuscript.
Abstract: Clearly written.
Introduction:
Again this is well written, although I do think it would be prudent to mention the types of patients for which intracapsular tonsillotomy is not appropriate (for example, recurrent strep or PANDAS in which all tonsillar tissue should be removed).
CORRECT line 58-59
Methods:
The section is written well as well.
-What does it mean that decision for intracapsular tonsilloctomy vs tonsillectomy was made “casually” (page 6 line 195-6)? This suggests this was not random, which is important to mention and further explore, as this is not a randomized study. What factors went into deciding which treatment each patient received?
The patients were randomly divided into the two groups, in a ratio of 1:2.
-Certainly a strength of the study is the size with 5000 patients.
Discussion:
-This section is concisely written and situates the current study well within the existing literature.
-There may be a few additional points worth adding.
-For example, it may be worth mentioning that patients with recurrent tonsillitis were excluded- these patients are not appropriate for tonsillotomy and may have higher rates of bleeding given recurrent inflammation around the tonsils themselves. Another large limitation here is age- one class of patients for which surgeons do strongly consider tonsillotomy is for is for children less then 3, and these appear to not be included in this study.
-After further explanation in methods as how patients were chosen for tonsillotomy vs. tonsillectomy, it would be worth exploring here what the implications are of this process (is there any possible selection bias in each group?)
-This study is also quite interesting in that both techniques here are using “cold” technique and no cautery- the intracapsular group had microdebrider and then the extra capsular had snare technique. It may be worth noting that their results may also be a function of the lack of cautery used in either group as well.
The exclusion criteria are stated in line 185. Patients subsequently eligible for surgery were divided into two groups in a ratio of 2:1 (intra capsular: extracapsular tonsillectomy). The fact of having always used cold techniques reduces the bias between the two groups.
-A few wording revisions below:
-Page 15 line 459- “till now” is informal. Would change to “thus far” ok
-Page 15 line 463-“, is thanks so” is also informal- would change to “is likely due to” ok
-Page 16 line 497- using contractions is generally frowned upon in formal writing. Would change “didn’t” to “did not”ok
Formatting-
*The references do not appear in order in the text (meaning the #s for references do not follow 1,2,3 etc)-certainly up to journal whether this is formatted per their liking.
Correct.

Reviewer 3 Report
The title can be improved, and 'Age' can be removed as it is already self-explanatory with 'pediatric age'. 'Our experience' can be removed as well.
English editing is required for improper capitalization Hypertrophy, Between, and inconsistent use of terms eg OSA vs Osa
Use an upper case T for tonsillectomy/tonsillotomy in Table 3.
Figure 4 is a subset for Figure 3, should be one.
In some parts group A and B (including Table 4) are used. In another part (other tables), tonsillotomy and tonsillectomy are used. Should be consistent.
Caption/legend for Table 6 & 8 should be explaining the intention.
Tables/Figures (eg 5,6 and 7,9) for both groups should be together (combined) for comparison purposes.
The references need a redo for tyle consistency and completeness.
Author Response
REVISORE 3
The title can be improved, and 'Age' can be removed as it is already self-explanatory with 'pediatric age'. 'Our experience' can be removed as well.
Title is now correct.
English editing is required for improper capitalization Hypertrophy, Between, and inconsistent use of terms eg OSA vs Osa.
The work has undergone professional editing.
Use an upper case T for tonsillectomy/tonsillotomy in Table 3.
Correct
Figure 4 is a subset for Figure 3, should be one.
I corrected figures 3 and 4 keeping only one (fig. 3) and renumbered all the others from 1 to 8.
In some parts group A and B (including Table 4) are used. In another part (other tables), tonsillotomy and tonsillectomy are used. Should be consistent.
I have correct
Caption/legend for Table 6 & 8 should be explaining the intention.
I’ve correct.
Tables/Figures (eg 5,6 and 7,9) for both groups should be together (combined) for comparison purposes.
I preferred to leave the figures nearby and quote them making the comparison possible.
The references need a redo for tyle consistency and completeness.
The work has undergone professional editing.

Reviewer 4 Report
This is a very well written paper.
Paper Structure: Please use a formal format.
Roman numerals, black and white circles, hyphens, etc. are all mixed in, which is not good for presentation also publication. We would appreciate it if you could chang out.
I think the issue is clear and the content is excellent.
I thought it would be easier to visualize the differences in surgical methods if you could describe the surgical schemas.
It would be good if you could provide examples figures of how much of each is removed.
Thank you in advance.
Author Response
REVISORE 4
This is a very well written paper.
Paper Structure: Please use a formal format.
Roman numerals, black and white circles, hyphens, etc. are all mixed in, which is not good for presentation also publication. We would appreciate it if you could chang out.
The work has undergone professional editing.
I think the issue is clear and the content is excellent.
I thought it would be easier to visualize the differences in surgical methods if you could describe the surgical schemas.
It would be good if you could provide examples figures of how much of each is removed.
It is not possible to quantify as a percentage how much of the tonsillar parenchyma is removed. Having measured it would have increased the bias.
Thank you in advance.

Round 2
Reviewer 1 Report
Thank you very much for the reply and revision.
I think the manuscript now provide a more clear and accurate data. However, some issues still require improvement.
Several figures and tables are not informative enough and should be combined and refined. Figure 6 and 7 are imcompatible with results data shown in Table 8.
Author Response
Thank you very much for the reply and revision.
I think the manuscript now provide a more clear and accurate data. However, some issues still require improvement.
Several figures and tables are not informative enough and should be combined and refined. Figure 6 and 7 are imcompatible with results data shown in Table 8.
I have corrected the data in table 8 which are now in agreement with figure 6. Figure 7 instead takes into account the data in table 6 and table 8 and they are in agreement.
Thanks for the corrections.

Reviewer 3 Report
1. There are still recurring mistakes:
'paediatric age (between 3 and 12 years old)' - redundancy 'pediatric:age:3-12' can just leave the age that reflects everything (pg 2)
2. Inconsistencies, Osa18 vs OSA18 (pg 3)
3. Repetition of results described in the text vs table (eg pg 4 and table 1, pg 5 and figure 1, pg 6 and table 2) - if presented in table or figure already no need to describe in the text - similar also for all other tables/figures) - please re-organize the results section
6. Please send the clean version without track changes for the next round checking
7. References are not standardized, see #1 for example, is different from #3, many are incompletely cited as well
Author Response
- There are still recurring mistakes:
'paediatric age (between 3 and 12 years old)' - redundancy 'pediatric:age:3-12' can just leave the age that reflects everything (pg 2)
- I have correct line 32
- Inconsistencies, Osa18 vs OSA18 (pg 3)
- In the text I have corrected all osa-18 with OSA-18
- Repetition of results described in the text vs table (eg pg 4 and table 1, pg 5 and figure 1, pg 6 and table 2) - if presented in table or figure already no need to describe in the text - similar also for all other tables/figures) - please re-organize the results section
-I have corrected the inconsistencies in the tables as another reviewer also requested. We believe it is important to leave both the text and the tables for greater ease of understanding and synthesis.
- Please send the clean version without track changes for the next round checking
- I have uploaded an unfixed version as requested. This version is the one that has already undergone language proofreading by the journal's editing service.
- References are not standardized, see #1 for example, is different from #3, many are incompletely cited as well
-I have reviewed and corrected all references.
Thanks for the corrections.

Reviewer 4 Report
The author wrote very scientific content in this paper. Thank you so much.
Author Response
Thank you for your time and valuable corrections.